# Short-Term Acute Exposure to Wildfire Smoke and Lung Function among Royal Canadian Mounted Police (RCMP) Officers

**DOI:** 10.3390/ijerph182211787

**Published:** 2021-11-10

**Authors:** Subhabrata Moitra, Ali Farshchi Tabrizi, Dina Fathy, Samineh Kamravaei, Noushin Miandashti, Linda Henderson, Fadi Khadour, Muhammad T. Naseem, Nicola Murgia, Lyle Melenka, Paige Lacy

**Affiliations:** 1Alberta Respiratory Centre, Department of Medicine, 559 Heritage Medical Research Centre, University of Alberta, Edmonton, AB T6G 2R3, Canada; moitra@ualberta.ca (S.M.); afarshch@ualberta.ca (A.F.T.); kamravae@ualberta.ca (S.K.); miandash@ualberta.ca (N.M.); l.melenka@synergyrespiratory.com (L.M.); 2Synergy Cardiac & Respiratory Care, Sherwood Park, AB T8H 0N2, Canada; dinafathy82@yahoo.com (D.F.); l.henderson@synergyrespiratory.com (L.H.); fkhadour@ualberta.ca (F.K.); naseem@ualberta.ca (M.T.N.); 3Department of Occupational and Environmental Medicine, University of Perugia, 06123 Perugia, Italy; nicola.murgia@unipg.it

**Keywords:** air pollution, lung function, occupational exposure, wildfire

## Abstract

The increasing incidence of extreme wildfire is becoming a concern for public health. Although long-term exposure to wildfire smoke is associated with respiratory illnesses, reports on the association between short-term occupational exposure to wildfire smoke and lung function remain scarce. In this cross-sectional study, we analyzed data from 218 Royal Canadian Mounted Police officers (mean age: 38 ± 9 years) deployed at the Fort McMurray wildfires in 2016. Individual exposure to air pollutants was calculated by integrating the duration of exposure with the air quality parameters obtained from the nearest air quality monitoring station during the phase of deployment. Lung function was measured using spirometry and body plethysmography. Association between exposure and lung function was examined using principal component linear regression analysis, adjusting for potential confounders. In our findings, the participants were predominantly male (71%). Mean forced expiratory volume in 1 s (FEV_1_), and residual volume (RV) were 76.5 ± 5.9 and 80.1 ± 19.5 (% predicted). A marginal association was observed between air pollution and higher RV [β: 1.55; 95% CI: −0.28 to 3.37 per interquartile change of air pollution index], but not with other lung function indices. The association between air pollution index and RV was significantly higher in participants who were screened within the first three months of deployment (2.80; 0.91 to 4.70) than those screened later (−0.28; −2.58 to 2.03), indicating a stronger effect of air pollution on peripheral airways. Acute short-term exposure to wildfire-associated air pollutants may impose subtle but clinically important deleterious respiratory effects, particularly in the peripheral airways.

## 1. Introduction

In the past three decades, the number of wildfire events in North America has increased significantly. There have been more than 120 major wildfire events in the United States and Canada between 1990 and 2020, out of which 41 took place in 2020 alone [1]. Between 2007 and 2017, over 6 million acres of land were burned every year in the US and Canada [2]. Rapid penetration of wildland areas for residential and industrial purposes, and climate change are some of the major reasons for the increasing number of wildland fires [3,4,5]. In May 2016, a major wildfire outbreak took place at Fort McMurray in the northern part of Alberta province in Canada, which led to the largest evacuation in Canadian history of over 80,000 people, and is considered the most expensive natural disaster in Canadian history with insured losses worth $3.7 billion [6]. This outbreak burned an area of 5890 km^2^, an area greater than the size of Prince Edward Island and affected approximately 8% of all private households in that region [6]. 

Wildfires severely impact the environment and human health. Wildfire smoke contains a wide range of gaseous compounds, such as carbon monoxide (CO), various oxides of nitrogen (nitric oxide, NO; nitrogen dioxide, NO_2_; and other oxides, NO_x_), and sulfur (sulfur dioxide, SO_2_; and other oxides, SO_x_), ozone (O_3_), methane (CH_4_), and many other polycyclic aromatic hydrocarbons (PAH), as well as the particulate matter of varying aerodynamic diameters (PMs) [7,8,9,10,11]. A higher emission of CO and CH_4_, and NO_x_ during a wildfire event facilitates tropospheric O_3_ production [11,12,13]. The gaseous and particulate matters produced by the wildfire smoke is more complex than what is found in vehicular exhaust, and its complexity also depends on various environmental aspects, such as the landscape of the burn area, seasonal conditions, and nature and phase of combustion (such as flaming and smoldering) [14,15]. Most importantly, the nature of different chemical production is dependent on the weather; for example, the generation of PMs in wildfire smoke is dependent on the condition (dry or wet) of the burning biomass and phase of combustion [15,16]. 

Most studies describe PMs as the most important wildfire pollutant to impact human health [15,16,17,18,19,20,21]. Exposure to wildfire-related PMs has been consistently shown to associate with increased inflammatory response and respiratory symptoms [15,21,22,23,24,25,26], emergency room visits and hospitalization [27,28,29], and mortality [21]. While there is consistent evidence of long-term exposure to PMs from wildfire smoke and respiratory outcomes, there is a dearth of evidence on the association between cumulative exposure to different wildfire-related pollutants and lung function. Understanding the clinical nature of respiratory function resulting from such exposure is important for the diagnosis of acute or chronic respiratory health events. Therefore, we aimed to study the association of lung function in relation to short-term acute exposure to different wildfire-related pollutants in first responder Royal Canadian Mounted Police (RCMP) officers during the Fort McMurray wildfire in 2016. 

## 2. Materials and Methods

### 2.1. Study Design and Participants

In this cross-sectional study, we investigated a group of RCMP officers who were deployed for the evacuation and rescue of people during a major wildfire that broke out in 2016 at Fort McMurray, located in the northern part of the province of Alberta in Canada. 218 officers were screened at Synergy Respiratory & Cardiac Care, Sherwood Park, Alberta. As this study was part of a surveillance program, no participants were excluded from the study. The officers visited the screening center over two years after their deployment. This study was performed according to the Declaration of Helsinki and was approved by the Health Research Ethics Board of Alberta (HREBA) (HREBA.CHC-18-0038) and Health Research Ethics Board, the University of Alberta (Pro00088553).

### 2.2. Demographic, Job, and Health-Related Information

An interviewer-administered structured questionnaire was used to capture information about the demographic profile (age, sex, smoking history, and frequency of smoking), and personal and family history (exposure to smoke at childhood, and parental lung disease). Details of the questionnaire have been described elsewhere [30]. Additionally, job exposure information including the dates and duration of deployment at the wildfire sites and use of respiratory protection (yes/no) were also recorded. Asthma was either self-reported or previously diagnosed by a physician or diagnosed at the clinic as per the guidelines [31]. 

### 2.3. Exposure Assessment

Information about the air quality indices associated with the Fort McMurray wildfire was obtained from the Athabasca valley air monitoring station, the nearest air quality monitoring station (~2 km) where all the officers were deployed. The Athabasca valley air monitoring station continuously (hourly) measures several indices of air quality along with weather conditions such as air temperature, relative humidity, wind speed and direction, and barometric pressure. Details of the instrumentation and measurement procedures are available elsewhere [32]. All the officers were stationed at Fort McMurray wildfire sites at different time points between 01 May and 31 May and were exposed continuously for at least 36 h. We accessed the air quality data from 01 May until 31 May 2016. Daily average concentrations of CO, CH_4_, NO, NO_2_, O_3_, SO_2_, and particulate matter with aerodynamic diameter ≤2.5 μm (PM_2.5_) were considered as the main air pollutants. 

We then calculated cumulative individual exposure to each of the air pollutants according to the formula developed previously [33], which can be expressed mathematically as the following:(1)$$E=\int_{t1}^{t2}C\left(t\right)dt$$
where, *E* is the total exposure to each of the pollutants; *C* is the concentration of the pollutant on day *t*, and was integrated from *t*1 to *t*2 (number of days spent at the wildfire sites). *t*1 and *t*2 are the first and last days of deployment. *C*(*t*) is the daily average of air pollutants.

### 2.4. Lung Function

A complete lung function profiling was performed for each of the officers. Spirometry and body plethysmography were performed using a Vmax^®^ Encore pulmonary function test system (Vyaire Medical, Mettawa, IL, USA) according to the American Thoracic Society/European Respiratory Society (ATS/ERS) guidelines for spirometry [34] and body plethysmography [35]. Forced expiratory volume in 1 s (FEV_1_), forced vital capacity (FVC), and the ratio of FEV_1_ and FVC (FEV_1_/FVC) were considered for this study. Percent of predicted values of these indices were calculated from the Canadian Cohort of Obstructive Lung Diseases (CanCOLD) reference equations of spirometry [36]. Total lung capacity (TLC), residual volume (RV), and their ratio (RV/TLC) were considered as the main plethysmo-graphic indices. Percent of predicted values for plethysmo-graphic indices were calculated from previously established reference equations for the Canadian population [37]. COPD was confirmed if the post-bronchodilator FEV1/FVC was lower than 0.7 [38].

### 2.5. Statistical Analyses

Variables were presented as mean (standard deviation, SD) or median (IQR) for continuous variables, and frequency (%) for categorical variables. At first, we analyzed the relationships between the exposure variables (individual exposure to the studied air pollutants) using Spearman’s rank-order correlation, as the variables were non-normally distributed. Secondly, we checked for collinearity between the air pollutant variables using variance inflation factor (VIF). As the variables were highly correlated (Spearman’s ρ range: 0.62 to 0.98; all *p*-value < 0.001) (Figure 1) and demonstrated very high collinearity (mean VIF = 158.7), we used a dimension reduction technique (principal component analysis, PCA) with a varimax rotation. Based on Eigenvalue > 1.0 (Appendix A), one principal component (PC1) was retained that explained 88% of the variance (Appendix A). We have used the term “air pollution index” for the principal component (PC1) in all subsequent analyses and further text.

To study the association between the air pollution index and lung function, we created a univariable (unadjusted) and a multivariable linear regression model for each of the lung function variables, taking into account potential confounders. Age, sex, body mass index (BMI), race (Caucasian vs. others), smoking history (never vs. ever smoker), and interval (days between deployment and screening) were tested as potential confounders and added into the multivariate models in a step-forward and backward approach and retained in the model (i) either based on an a priori evidence, or (ii) if the covariate influenced the estimates of the remaining variables by more than 10%. Finally, we considered age, sex, BMI, race, and smoking history as confounders in the adjusted models. Heteroskedasticity of the models was checked using Cook-Weisberg’s test [39]. The goodness of fit of the models was assessed using Akaike’s information criteria (AIC) [40]. 

In addition, we also performed several secondary analyses. Firstly, we adjusted the multivariate models additionally for the use of personal protective equipment (PPE). Secondly, based on a priori evidence of occupational irritants exposure-associated reactive airways dysfunction syndrome (RADS) that can persist for >3 months [41], we stratified the multivariable models by interval (≤90 days and >90 days) to test the short-term and long-term association between air pollution index and lung function and the estimates were compared using the Chow test [42]. Lastly, we tested potential effect modification by smoking (never vs. ever smoker), presence of a diagnosed airway disease (asthma or COPD), exposure to second-hand smoke in childhood, and parental lung disease. All analyses were performed using a complete case approach in Stata V.15.1 (StataCorp, College Station, TX, USA). An alpha level of 0.05 was considered as the threshold for statistical significance. 

## 3. Results

### 3.1. Study Population Characteristics and Air Pollution Exposure

Of those screened, most participants were Caucasian (96%) and male (71%) with a mean age of 38 (standard deviation, SD: 9) years (Table 1). 81% were never-smokers. The median exposure duration of the participants was 8 (interquartile range, IQR: 7, 10; min, max: 1, 28) days. The Median (IQR) interval time was 60 days (22, 627 days). Participants were exposed to a very high number of carboniferous compounds (such as CO and CH_4_) (ranging between 9680 and 25,945 μg/m^3^) in addition to a high amount of exposure to O_3_ and PM_2.5_ (ranging between 447 and 2143 μg/m^3^) during the entire period of deployment. Mean (SD) FEV_1_ (% predicted), RV (% predicted), and RV/TLC of the participants were 96.2 (12.4), 80.1 (19.5), and 22.4 (4.8), respectively.

### 3.2. Association between Air Pollution and Lung Function

In the unadjusted models, we did not observe any significant association between the air pollution index and spiro-metric lung function variables (FVC, FEV_1_, and FEV_1_/FVC). However, we observed that the air pollution index was marginally associated with peripheral airway dysfunction as measured by plethysmography, i.e., an IQR increment of air pollution index was associated with an increase in RV (% predicted) (β: 1.76; 95% confidence interval (CI): −0.06 to 3.57) and RV/TLC (0.40; −0.05 to 0.85) (Figure 2). In multivariable models adjusted for potential confounders, although the magnitude of the estimates was minimized, the directionality of the associations remained unchanged. Upon further adjusting the multivariable models by the use of PPE, we did not observe any change in the estimates of both spiro-metric and plethysmo-graphic lung function variables (Appendix A).

However, after stratifying the associations by a specific interval (≤90 days and >90 days), we found that there was a significant adverse effect of air pollution on RV and RV/TLC among participants who were screened ≤90 days (*n* = 133) compared with those screened >90 days of deployment (*n* = 85) (Figure 3). For example, we observed a 2.8% increase in RV (95% CI: 0.91 to 4.70; *p* < 0.01) and a 0.59% increase in RV/TLC (95% CI: 0.06, 1.10) per IQR increment of air pollution index in the first group, whereas no such effect was observed in the latter group. Despite these changes, effects of air pollution were not observed for spiro-metric lung volumes. We did not observe any effect modification by smoking, presence of any prior airway obstruction due to existing illness (asthma and COPD), exposure to second-hand smoke in childhood, or any parental history of lung disease (Appendix A).

## 4. Discussion

In this study, we found RCMP officers deployed during the 2016 Fort McMurray wildfires were exposed to an exceptionally high level of wildfire-related air pollutants (primarily to CO, CH_4_, NO_2_, O_3_, and PM_2.5_) over a short period. In our analysis, although short-term acute exposure to wildfire-related air pollutants (denoted as “air pollution index” in the text) was not associated with spiro-metric lung function (FEV_1_, FVC, FEV1/FVC), we observed a marginal association with small airway functions as observed in the plethysmo-graphic recording. A higher association between air pollution index and small airway (indicated by higher RV and RV/TLC) was observed in those participants who were screened within the first 90 days of deployment. However, no such association was found in those who were screened later (90 days post-deployment). We did not observe any effect modification by smoking, any pre-existing airway obstruction due to asthma or COPD, childhood smoke exposure, or any parental history of lung disease.

We did not test the association between each of the air pollutants and lung function separately; nevertheless, our data suggest that the participants were primarily exposed to exceptionally high concentrations of a mixture of different air pollutants that appeared to have a combined effect on small airway function. Although the literature on the effect of CO on lung function is scarce, some studies have demonstrated lung function decline in association with environmental exposure to CO, particularly in children [43] and asthmatics [44]. Another recent study demonstrated that kitchen stoves using liquefied petroleum gas (LPG) also emit CO, which is associated with lower lung function [45]. Moreover, the effect of CO may be more profound in the presence of high levels of carbon dioxide (CO_2_), although the concentration of CO_2_ at wildfire sites could not be obtained. 

Participants in our study were also exposed to very high amounts of CH_4_. Although methane-induced respiratory problems are not well studied, one case report suggested that CH_4_ inhalation was associated with acute pneumonitis [46]. Despite some epidemiological evidence for CH_4_ exposure-associated respiratory conditions [47,48], how this molecule affects lung physiology is not well understood. Although an independent assessment between CH_4_ and lung function is not practically feasible in this study, we may postulate based on previous literature that CH_4_ may affect lung function in the presence of other pollutants and oxidizing agents, such as O_3_. 

The effect of O_3_ on lung function has been studied extensively [49,50,51,52,53]. It is evident that higher exposure to O_3_ is associated with reduced small airway function [50,51,52], and this mechanism is mediated by neutrophilic inflammation as well as other pro-inflammatory responses in the distal airways, which serve to increase airway resistance in the peripheral lungs [50,54]. All these previous reports support our findings of lowered small airway function in association with exposure to air pollutants. 

NO_2_ and PM_2.5_ are generated during the combustion of biomass and fossil fuels and are considered the most important air pollutants in causing respiratory damage. The damaging effects of NO_2_ on small airway function were first established more than 40 years ago. Inhalation of NO_2_ may induce bronchiolitis obliterans, a typical feature of small airway damage [55]. Such manifestations might develop within 2 weeks of exposure but recover over time [55]. This further supports our observation of elevated small airway obstruction in those who were screened early after exposure. Several other studies have reinforced evidence of the injurious effects of NO_2_ on lung function, particularly on small airways [56,57]. However, it has been noted that NO_2_ and PM_2.5_ usually coexist, and the effect of NO_2_ on the lungs is enhanced by PM_2.5_ [56]. PM_2.5_ contains ultrafine particles that can easily penetrate the distal parts of the lungs, deposit in the alveoli, and can also cross the blood-gas barrier [58]. Studies have revealed that exposure to ultrafine particulate matter may elicit acute lung function changes, particularly in the small airways [59]. Our observation of small airway changes in association with a synergistic effect of PM_2.5_ and other hazardous gases is also substantiated by other studies describing the impact of PM_2.5_ on small airway function [56,60]. It must be remembered that gases that have low solubilities, such as oxides of nitrogen and sulfur and particulate matters of very small aerodynamic diameter (PM_2.5_), can reach up to the alveoli and terminal bronchioles without affecting the central airways to an observable extent [61]. This supports our findings of a significant association between higher exposure to air pollution and lower RV but no other lung function variables. Although our observation of significant alteration of RV only in individuals examined within 90 days of deployment indicates a probable temporary effect on the distal airways, it is to be noted that chronic exposure to such air pollutants may induce acute inflammation in the distal airways and enhance structural changes in these areas [62] that may lead to small airway remodeling, a gateway of chronic airflow obstruction. However, we could not confirm the underlying mechanisms, but we assume that these air pollutants. Nevertheless, these changes are often subtle and are not easily detected by spirometry.

Our findings provide clinical insight into respiratory health effects associated with short-term acute exposure to air pollutants. The results endorse the use of high-sensitivity lung function tests such as body plethysmography and potentially oscillometry over conventional spirometry for the detection of subtle yet clinically important changes in the lungs due to exposure to air pollutants, which may not be detected in spirometry. This also proposes that the peripheral airways are more responsive to any exogenous triggers, which is reflected by the minimal yet clinically volumetric change in RV important (for example, we observed ~3% increase of RV (% predicted) in relation to per IQR change in air pollution index among those screened within 90 days of deployment), but not in the proximal airways. We also vouch for adequate respiratory protection for first responders involved in fire control who are likely to be exposed to very high levels of gaseous and particulate matters. Moreover, we found a much higher prevalence of asthma among the officers than the average Canadian population [63]. Although we could not diagnose any events of work-related asthma in our participants, the alarmingly higher prevalence of asthma among the participants than that of the average Canadian population underscores plausible work-related exposures, which may be responsible for respiratory illnesses. Therefore, our study reinstates the need for a holistic assessment of occupational and environmental exposure along with state-of-the-art clinical investigations for a comprehensive diagnosis of the health conditions. Lastly, proper monitoring and surveillance of any potential workplace exposures are strongly recommended.

Our study offers several new pieces of information to the existing knowledge of wildfire smoke exposure and respiratory health effects. To the best of our knowledge, this is the first study to demonstrate an association between short-term acute exposure to wildfire smoke and small airway function. Secondly, unlike other wildfire-related studies that have mostly focused on particulate matter, we considered both particulate and gaseous pollutants that enabled us to estimate the cumulative effects of all the pollutants. Lastly, while other studies use only spirometry for lung function measurement and did not pay attention to static lung volumes or air trapping, we used both spirometry and body plethysmography for a more comprehensive assessment of the respiratory health of the participants, particularly with a focus on the small airways.

However, our study has some limitations. This is a cross-sectional study; therefore, a causal relationship between exposure to air pollutants and lung function changes cannot be established, although our results are in line with previously established epidemiological evidence and plausible biological effects of air pollution on lung function changes. Moreover, we did not have any health-related data of the participants before their exposure. Thus, a longitudinal or follow-up study design to estimate the effect of the exposure on their lung health could not be performed. Second, in this study, all the participants were exposed to high levels of pollutants. Therefore, we could not compare their respiratory health effects with low-exposed individuals. Third, the study was not designed prospectively and was a result of an unplanned health surveillance program followed by a natural disaster, and the participants were not recruited through proper inclusion-exclusion criteria. Therefore, we had a relatively smaller sample size than many other properly designed epidemiological studies of this nature; thus, an inadequate statistical power could be another limitation of this study. Fourth, we could not perform any hematological or immunological profiling, which could delineate any underlying acute or chronic inflammation. Fifth, although we calculated individual exposure to air pollutants indirectly, a direct approach, i.e., by using a personal air sampler, would more accurately measure exposures and specific deployment sites of participants. Last, we did not measure airway resistance, which could provide us with more clinically important information about any structural changes in the peripheral airways. 

## 5. Conclusions

We found that short-term acute exposure to wildfire-related air pollutants was marginally associated with lowered small airway function and that these subtle changes were not reflected in spirometry. Our results also suggest that such short-term exposure to air pollutants may cause changes in the distal parts of the lungs, which need to be detected at an early stage. To the best of our knowledge, this is the first study to investigate small airway function due to a short-term acute wildfire-related smoke exposure, taking into consideration a wide range of gaseous and particulate matters. Our results provide further substantiate previously published findings linking air pollution with lung function changes, particularly, with small airways, and call for more advanced approaches for an early diagnosis of respiratory conditions.

## Figures and Tables

**Figure 1 ijerph-18-11787-f001:**
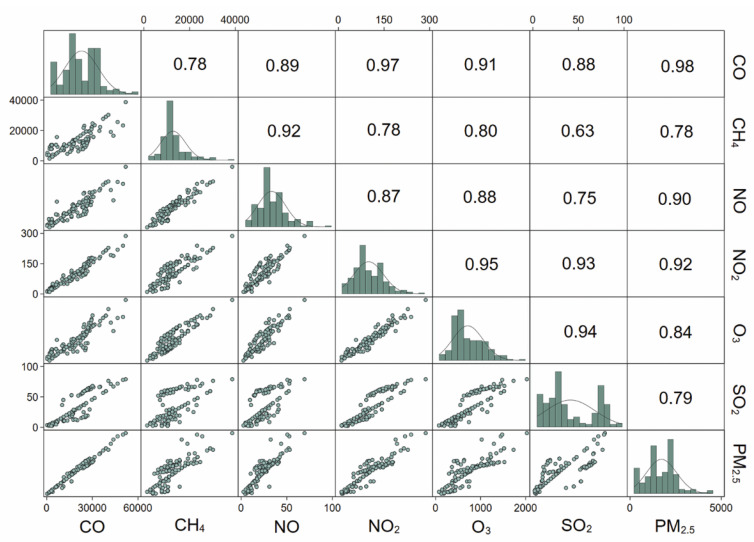
Scatterplot matrix and distribution (histograms) of continuous air pollution indices. Bivariate scatterplots of continuous variables below the diagonal; (distribution) histograms of each variable on the diagonal; Spearman’s correlation coefficient (ρ) above the diagonal. Note: All correlation values are significant at *p* < 0.001. Abbreviations: CO: carbon monoxide; CH_4_: methane; NO: nitric oxide; NO_2_: nitrogen dioxide; O_3_: ozone; SO_2_: sulfur dioxide; PM_2.5_: particulate matter ≤ 2.5 μm in diameter. All units are in μg/m^3^.

**Figure 2 ijerph-18-11787-f002:**
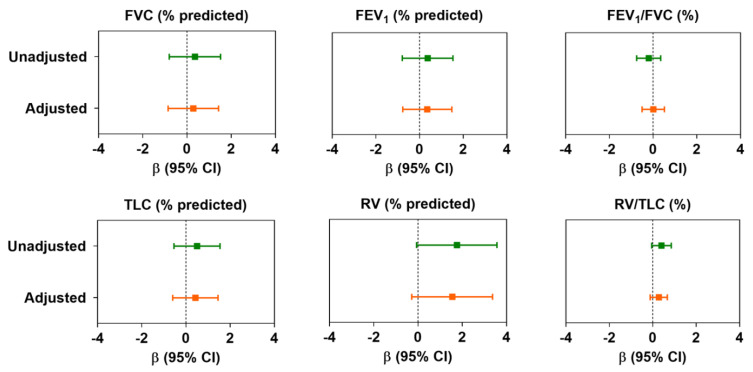
Changes in lung function in association with per interquartile (IQR) change in air pollution index. Data presented as linear regression coefficient (β) and 95% confidence interval, unless otherwise specified. Coefficients were calculated with respect to per IQR change (μg/m^3^) of air pollution index (principal component). In adjusted models, age, sex, BMI, race, and smoking history were considered as confounders. For abbreviations, see text.

**Figure 3 ijerph-18-11787-f003:**
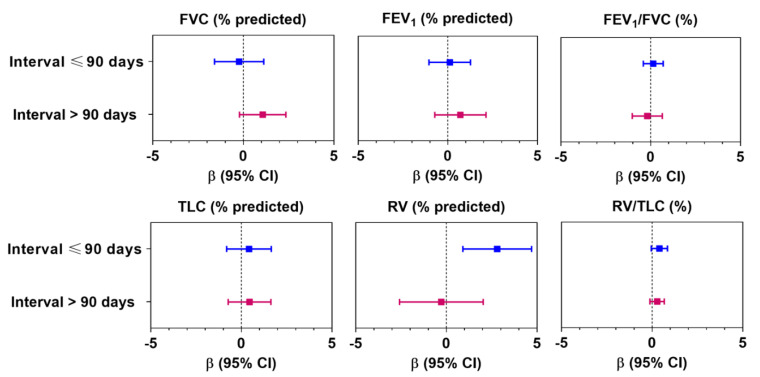
Association between changes in lung function in association with per interquartile (IQR) change in air pollution index stratifying by interval (days between deployment and screening). Data presented as linear regression coefficient (β) and 95% confidence interval, unless otherwise specified. Coefficients were calculated with respect to per IQR change (μg/m^3^) of the air pollution index (principal component). In adjusted models, age, sex, BMI, race, and smoking history were considered as confounders. For abbreviations, see text.

**Table 1 ijerph-18-11787-t001:** Demographic characteristics, exposure history, and clinical profiles of the RCMP officers.

**Demographics**	***N* = 218**
Sex (male), *n* (%)	155 (71)
Age (years), mean (SD)	38 (9)
BMI (kg/m^2^), mean (SD)	29.8 (5.2)
Ethnicity, *n* (%)	
Caucasian	209 (96)
Others	9 (4)
Never smokers, *n* (%)	176 (81)
Passively smoke exposure at childhood, *n* (%)	131 (60)
Parental lung disease, *n* (%)	33 (15)
Family history of cancer, *n* (%)	49 (23)
Personal PPE used while deployed, *n* (%)	147 (68)
Days spent amid wildfire, median (IQR)	8 (7, 10)
Interval (days), median (IQR) ^†^	60 (3, 627)
Asthma, *n* (%)	65 (30)
COPD, *n* (%)	5 (2)
**Personal exposure details**	
CO (μg/m^3^), median (IQR)	17,386.8 (11,509.3, 25,945.0)
CH_4_ (μg/m^3^), median (IQR)	11,063.6 (9680.7, 13,413.1)
NO (μg/m^3^), median (IQR)	19.9 (16.9, 28.8)
NO_2_ (μg/m^3^), median (IQR)	90.3 (68.8, 140.0)
O_3_ (μg/m^3^), median (IQR)	590.8 (446.8, 992.1)
SO_2_ (μg/m^3^), median (IQR)	22.6 (14.6, 57.8)
PM_2.5_ (μg/m^3^), median (IQR)	1632.9 (1014.3, 2142.6)
**Lung function**	
FEV_1_ (% predicted), mean (SD)	96.2 (12.4)
FVC (% predicted), mean (SD)	100.8 (12.3)
FEV_1_/FVC (%), mean (SD)	76.5 (5.9)
TLC (% predicted), mean (SD)	95.3 (11.1)
RV (% predicted), mean (SD)	80.1 (19.5)
RV/TLC (%), mean (SD)	22.4 (4.8)

Data presented as frequency (%), mean (standard deviation: SD), or median (interquartile range: IQR), unless otherwise specified. ^†^ Interval: days between deployment and screening. Abbreviations: BMI: body mass index; COPD: chronic obstructive pulmonary disease; FEV_1_: forced expiratory volume in 1 s; FVC: forced vital capacity; PPE: personal protective equipment.

## Data Availability

Data of this manuscript may contain sensitive personal information, and share, transfer, or modification of the data is prohibited by law. However, an anonymous dataset with limited information can be shared upon reasonable request made to the corresponding author.

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
