# Peer review of "Short-Term Acute Exposure to Wildfire Smoke and Lung Function among Royal Canadian Mounted Police (RCMP) Officers"

_ijerph, 2021, doi:10.3390/ijerph182211787_

Round 1

Reviewer 1 Report

  1. The authors used cumulative exposures (i.e., concentration times duration of exposure). My concern is whether the duration of the deployment would be affected by adverse health outcomes experienced by the officers as a result of the wildfire exposures. For example, if, because of the wildfire exposures, officers who developed respiratory symptoms ended the deployment sooner, while officers who didn't have respiratory reactions stayed on the deployment for a longer time, then it would appear that people with higher cumulative exposures had better health outcomes; if this is the case, negative bias will occur (i.e., the estimated effect of wildfire would be lower than the true). This may not have too large an impact given that the duration of the deployment was relatively similar across the officers (IQR of 7-10 days), but I still wonder if the authors have collected information about the officers' reasons for ending the deployment (e.g., normal or due to medical concerns); and it is helpful to also provide the range (min and max) of deployment duration. The authors may also conduct a sensitivity analysis using exposure metrics that are not related to the duration of the deployment, say the average or maximum concentrations during the deployment, to see if the pattern of the results stays the same. 
  2. The main estimate of this paper is that RV increased 1.76 (or 2.8 in a sub-analysis among officers with <=90 days intervals) per IQR increase in the exposure index. The authors may explain or add some discussion on the clinical relevance/meaning of these estimates (what does an increase of 1.76 or 2.8 mean, clinically). It would be helpful to cite and compare with studies that also examined RV as a health endpoint for air pollution exposures. 
  3. In the Methods section, after explaining the rationale for covariate selection, please also mention which covariates were included in the final models and how they were specified (continuous or categorical). I noticed that the selected covariates only appeared in the footnote of the Figures. 

Author Response

  1. The authors used cumulative exposures (i.e., concentration times duration of exposure). My concern is whether the duration of the deployment would be affected by adverse health outcomes experienced by the officers as a result of the wildfire exposures. For example, if, because of the wildfire exposures, officers who developed respiratory symptoms ended the deployment sooner, while officers who didn't have respiratory reactions stayed on the deployment for a longer time, then it would appear that people with higher cumulative exposures had better health outcomes; if this is the case, negative bias will occur (i.e., the estimated effect of wildfire would be lower than the true). This may not have too large an impact given that the duration of the deployment was relatively similar across the officers (IQR of 7-10 days), but I still wonder if the authors have collected information about the officers' reasons for ending the deployment (e.g., normal or due to medical concerns); and it is helpful to also provide the range (min and max) of deployment duration. The authors may also conduct a sensitivity analysis using exposure metrics that are not related to the duration of the deployment, say the average or maximum concentrations during the deployment, to see if the pattern of the results stays the same.

Re:  Thank you very much for your suggestions. We have mentioned the range of deployment in the manuscript, as the following:

“The median exposure duration of the participants was 8 (interquartile range, IQR: 7, 10; min, max: 1, 28) days.” (Page: 4-5, line number: 174-175)

  • Regarding your query regarding the end of deployment, unfortunately we have no information on the reason why the duration of deployment varied among the officers. Therefore, we cannot comment whether the officers were called off due to routine operational procedure or due to any health problem.
  • Regarding your suggestion about using exposure matrix, this would be an excellent sensitivity analysis. Unfortunately, we did not have any information about their exposures other than the studied exposures which were also dependent on their deployment duration. Thus, we think an exposure matrix without deployment status may be difficult to apply and may not provide an accurate estimation.

  1. The main estimate of this paper is that RV increased 1.76 (or 2.8 in a sub-analysis among officers with <=90 days intervals) per IQR increase in the exposure index. The authors may explain or add some discussion on the clinical relevance/meaning of these estimates (what does an increase of 1.76 or 2.8 mean, clinically). It would be helpful to cite and compare with studies that also examined RV as a health endpoint for air pollution exposures.

Re: Thank you for your suggestion. We have added a line in the discussion section for better understanding, which reads as the following:

“This also proposes that the peripheral airways are more responsive to any exogenous triggers, which is reflected by the minimal yet clinically volumetric change in RV important (for example, we observed ~3% increase of RV (% predicted) in relation to per IQR change in air pollution index among those screened within 90 days of deployment), but not in the proximal airways.” (Page: 8, line: 308-311)

However, we did not find any previous literature emphasizing RV as the endpoint for air pollution exposure. This is presumably because most of the reports dealing with air pollution primarily focussed on FEV1 and FVC and did not pay attention to static lung volumes or air trapping, and thus did not focus on peripheral/small airways. In that sense, our study is the first to evaluate both central and peripheral airways in relation to air pollution exposure that we have already mentioned in the novelty section (please see Page 9, line: 319-320) as the following:

"Lastly, while other studies use only spirometry for lung function measurement and did not pay attention to static lung volumes or air trapping, we used both spirometry and body plethysmography for a more comprehensive assessment of the respiratory health of the participants, particularly with a focus on the small airways."

  1. In the Methods section, after explaining the rationale for covariate selection, please also mention which covariates were included in the final models and how they were specified (continuous or categorical). I noticed that the selected covariates only appeared in the footnote of the Figures.

Re: Thank you very much for the suggestion. We have now mentioned the covariate selection in the text that reads as the following:

“Finally, we considered age, sex, BMI, race, and smoking history as confounders in the adjusted models.” (Page: 4, line: 158-159)

Reviewer 2 Report

Dear Editor

I thank for giving this opportunity to review this article titled "Short-term acute exposure to wildfire smoke and lung function 2
among Royal Canadian Mounted Police (RCMP) officers"

I appreciate authors for a brief with appropriate citations.

This is an interesting study to investigate small 
airway function due to a short-term acute wildfire-related smoke exposure, taking into consideration a wide range of gaseous and particulate matters.

They have shown that such short-term exposure to air pollutants may cause changes in the distal parts of the lungs, which need to be detected at an early stage.

Comment: Why did authors choose 90 days window for the analysis?

Author Response

Why did authors choose 90 days window for the analysis?

Re: Thank you for your query. We apologize that the principle behind the stratification was not very well described in the manuscript. However, we have now incorporated the reason behind the stratification strategy, i.e., ≤90 days and > 90 days as the following:

“Secondly, based on a priori evidence of occupational irritants exposure-associated reactive airways dysfunction syndrome (RADS), that can persist for >3 months, we stratified the multivariable models by interval (≤90 days and >90 days) to test the short-term and long-term association between air pollution index and lung function and the estimates were compared using the Chow test [42]” (Page: 4, line: 164-169]

Reviewer 3 Report

Estimated Authors of the paper "Short-term acute exposure to wildfire smoke and lung function among Royal Canadian Mounted Police (RCMP) officers", 

I've read your contribution to IJERPH with great interest. In your cross-sectional study, you assessed how the occupational exposure to wildfires impacted on the respiratory function test of Mounted Police Officers. In fact, your study suggests that the effect, if any, was limited to the small airways, with seemingly no significant long-term impact on the respiratory function tests.

In other words, the present research is both interesting "per se" and consistent with the aims of the present journal.

However, I'm recommending some minor revisions of the present paper for the following reasons:

1) methods section should be improved clearly explaining what the interval time is; in fact, the text is incomplete and forces the reader to the caption of table 1. In this regard, methods should more precisely explain that you stratified your participants in two groups, with interval in screening <= 90 vs. > 90 days from the initial exposure (at the moment, it is explained in results section rows 209-210); the number of individuals from each group should be reported in the Table 1.

2) I've some concerns regarding the effects reported in Figure 3 and discussed accordingly. According to the Figure 3, the lower limit of RV/TLC% < 90 days intercepts the 0 limit, therefore a significant effect should be more cautiosly reported and discussed. Otherwise, please report more extensively (as you did for RV%: "... we observed a 2.8% increase in RV (95%CI: 0.91 to 4.70, p< 0.01") the effect you identified for RV/TLC%.

3) a minor but annoying issue is represented by the notation of chemical substances across the text. CH4 should be written as CH4, O3 as O3 etc.

Author Response

  1. Methods section should be improved clearly explaining what the interval time is; in fact, the text is incomplete and forces the reader to the caption of table 1. In this regard, methods should more precisely explain that you stratified your participants in two groups, with interval in screening <= 90 vs. > 90 days from the initial exposure (at the moment, it is explained in results section rows 209-210); the number of individuals from each group should be reported in the Table 1.

Re: Thank you very much for your constructive comments and suggestions. Your comment is the same of another reviewer and we sincerely apologize for not describing our stratification strategy very clearly in the manuscript. We have now added this in the revised manuscript that reads as the following:

“Secondly, based on a priori evidence of occupational irritants exposure-associated reactive airways dysfunction syndrome (RADS) that can persist for >3 months, we stratified the multivariable models by interval (≤90 days and >90 days) to test the short-term and long-term association between air pollution index and lung function and the estimates were compared using the Chow test [42]” (Page: 4, line: 164-169]

  1. I've some concerns regarding the effects reported in Figure 3 and discussed accordingly. According to the Figure 3, the lower limit of RV/TLC% < 90 days intercepts the 0 limit, therefore a significant effect should be more cautiously reported and discussed. Otherwise, please report more extensively (as you did for RV%: "... we observed a 2.8% increase in RV (95%CI: 0.91 to 4.70, p< 0.01") the effect you identified for RV/TLC%.

Re: We thank you for bringing this issue to our notice. We want to bring to your notice that the lower limit did not intercept 0 for RV/TLC% ≤90 days [β=0.59; 95%CI: 0.06, 1.10], but probably due to small bars, it appears differently. We, therefore, have now added the confidence interval in the result section [please see Page: 6, line: 217-218].

  1. a minor but annoying issue is represented by the notation of chemical substances across the text. CH4 should be written as CH4, O3 as O3 etc.

Re: Thank you very much for stating that. I was informed by the journal that these would be edited during typesetting. However, we have corrected the notation throughout to ensure consistency.